# Evaluation of an Online Version of the CFT 20-R in Third and Fourth Grade Children

**DOI:** 10.3390/children9040512

**Published:** 2022-04-04

**Authors:** Linda Visser, Josefine Rothe, Gerd Schulte-Körne, Marcus Hasselhorn

**Affiliations:** 1Department of Education and Human Development, DIPF | Leibniz Institute for Research and Information in Education, 60323 Frankfurt am Main, Germany; hasselhorn@dipf.de; 2Center for Research on Individual Development and Adaptive Education of Children at Risk (IDeA), 60323 Frankfurt am Main, Germany; 3Department of Child and Adolescent Psychiatry, Faculty of Medicine Carl Gustav Carus, Technische Universität Dresden, 01307 Dresden, Germany; josefine.rothe@uniklinikum-dresden.de; 4Department of Child and Adolescent Psychiatry, Psychosomatic and Psychotherapy, Ludwig-Maximilian-University Munich, 80336 Munich, Germany; gerd.schulte-koerne@med.uni-muenchen.de

**Keywords:** intelligence test, CFT 20-R, digitalisation, psychometric characteristics, equivalence

## Abstract

There is growing demand for digital intelligence testing. In the current study, we evaluated the validity of an online version of the revised German Culture Fair Intelligence Test (CFT 20-R). A total of 4100 children from the third and fourth grades completed the online version using a smartphone or tablet. Subsequently, 220 of these children also completed the paper-pencil (PP) version. The internal consistency and construct validity of the online version appeared to be acceptable. The correlation between the raw scores and school grades in German and mathematics was slightly lower than expected. On average, the raw scores for the PP version were revealed to be higher, which was probably due to a learning effect. At the item level, the results show small differences for the subtests Series and Matrices, possibly caused by small differences in the presentation of the items. The correspondence between the versions did not depend on children’s levels of impulsivity or intelligence. Altogether, the results support the hypothesis that the online version of the CFT 20-R is a valid measure of general fluid intelligence and highlight the need for separate norms.

## 1. Introduction

In the context of school, the assessment of intelligence plays an important role in diagnosing special needs. For example, when diagnosing special learning disabilities, below-average intelligence needs to be ruled out. According to the International Classification of Diseases [1], there needs to be a discrepancy between the learning performance and intelligence for the diagnosis of a specific learning disability to be given. This double discrepancy criterion has been debated in recent years, as most studies do not find differences between children who do and children who do not fulfil this criterion. Therefore, the Diagnostic Manual of Mental Disorders [2] does not apply this criterion, but instead contains an exclusion criterion for below-average intelligence [3]. In addition, information about a child’s intelligence is informative for planning suitable interventions and as a basis for advice about school placement. 

In today’s age of digitalisation, there is increasing demand for the assessment of intelligence in a digital format as an alternative to the traditional paper-pencil (PP) format. A digital format facilitates adaptive testing and standardisation during the test administration and makes the calculation of the test results quicker and less prone to errors in comparison to manual calculation.

### 1.1. Administration Mode Effects 

When an assessment instrument is being transformed from PP to a digital format, the content does not change. However, changes in the presentation of test items and task requirements cannot be ruled out. For example, items are often presented one by one instead of listwise, which means that it is no longer possible to change responses to items earlier in the test [4], possibly influencing the test result (item presentation effect). The changes in the test that result from a change in test format (mode effect) require an evaluation of the comparability of the test in its PP and digital formats [5]. Such mode effects have already been evaluated for various response formats (e.g., multiple choice or open) and competence areas (e.g., reading comprehension, nonverbal cognitive abilities; [6]). Meta-analytical results have shown that factors such as the competence area or test nature (linear or adaptive) can influence the strength and direction of modus effects, e.g., [7,8,9]. For example, the meta-analysis by Kingston [9] showed that for the assessment of math competencies, PP formats often entail slight advantages. In contrast, for the assessment of linguistic (English) competencies, a slight advantage of a digital test format could be identified. Other meta-analyses report larger performance differences between digital and PP test formats for both reading comprehension and mathematical competencies in linear compared to adaptive testing procedures [8,9]. An additional factor that influences performance differences between PP and digital test formats in various competence domains is the amount of experience with computers, e.g., [10,11]. More specifically, more experience appears related to relatively higher performance in digital test formats in pupils. However, the interpretation of such results is complicated by the rapid changes in both the digital assessments used and in children’s media use.

The reported results come from studies conducted about 20 years ago among 8th grade pupils and using local computers and laptops. More recently, the results of the Trends in International Mathematics and Science Study (TIMSS) showed no mode effect when comparing test results in PP and tablet format among Dutch fourth grade children [12]. The authors explain this result based on the generally high familiarity of the pupils with tablets. Based on an international sample of fourth grade children from 24 countries, a modus effect was found [13]. However, this study differed from the one by Hamhuis et al. [12], in that both tablets and PCs were used for digital testing and the results were not analysed separately. In addition, the operationalisation of familiarity with the testing device differed considerably between the studies. Altogether, the results support the need to evaluate test mode effects.

### 1.2. Interaction of Personality Traits and Mode Effects 

Research into test mode effects so far has not yet focused on the question to which extent the comparability of testing formats depends on particular characteristics of the children, such as their impulsivity. Possibly, children who generally respond impulsively do so to a lesser degree in a digital testing format, because they do not get distracted by other items [14]. When a digital assessment instrument is embedded in a child-friendly cover story with a reward system, motivational factors play a role as well. Such cover stories have been shown to have positive effects on the executive functions (e.g., inhibition) in persons with attention deficit hyperactivity disorder (ADHD; e.g., [15]), which in turn can positively influence the performance of items assessing complex abilities such as fluid intelligence, e.g., [16].

Impulsive testing behaviour is also reflected in the so-called speed–accuracy trade off. This means that, for tests with a time limit, a faster response pattern is related to lower accuracy and vice versa. In problem-solving tasks, there is a positive relationship between the time spent on a task and the accuracy of the response. This relationship is stronger with increasing task difficulty [17]. Apart from the speed-accuracy trade off, which depends on the test taker’s ability, test takers also differ in their tendency to focus more on speed or on accuracy. This speed–accuracy emphasis is a personal characteristic and has been shown to be independent of intelligence [18,19]. It has not yet been studied if differences exist between PP and digital test formats in speed accuracy emphasis.

### 1.3. Specifics of Online Assessments 

Online test formats are a special form of digital assessments. These can be performed at any location and using various devices (e.g., tablet, smartphone, PC). On the one hand, this yields flexibility. On the other hand, it can reduce the degree of standardisation in both the testing medium and the test environment. A change in test medium alone could have an influence on the test result. One factor that can differ between different test media is the input mode (keyboard, mouse, touchscreen), of which the influence has been thoroughly evaluated. It has been shown that selecting objects or response options via a touchpad is significantly faster than with a computer mouse, e.g., [20,21], which will affect the test results in case of speed tests. This effect appears to be independent of age. For example, Findlater et al. [21] showed that older persons (61–86 years) are slower, on average, when using a computer mouse or touchpad compared to younger persons (19–51 years). However, older persons benefited more from using a touchpad compared to younger persons (35% versus 16% faster response times compared to the computer mouse).

An additional characteristic of online assessments is that these often take place without observation and instruction by a trained person. Consequently, the test behaviour cannot be controlled. A meta-analysis [22] has shown that unobserved test situations are related to better test performance compared to observed test situations. This effect is larger for declarative knowledge (which can be looked up on the internet) than for other test domains, such as figural reasoning as measured by the Culture Fair Intelligence Test (CFT; [23]).

### 1.4. Online Version of the Culture Fair Intelligence Test

The current study focuses on the revised German Culture Fair Intelligence Test 2 (CFT 20-R; [14]), which is a language-free intelligence test that is widely used in German-speaking countries for assessing persons from 8.5 years onwards. It assesses the fluid intelligence and claims to be culturally fair in that test persons who are less familiar with the German culture and/or language are not disadvantaged. Fluid intelligence was first defined by Cattell [24,25], who divided intelligence into two dimensions: fluid and crystallised. Fluid intelligence refers to the ability to solve problems and adapt to new situations and is largely independent from cultural influences. Crystallised intelligence is the cumulative knowledge acquired by previous learning and is considered to depend on the culture.

The previous version, the CFT 20 [26], has been adapted to a PC version by the test authors. A comparison between the PP and PC versions with a sample of 94 ninth grade pupils showed no differences between the mean test results. Therefore, separate norms were not deemed necessary. However, only for children in special schools, the correlation between the two versions was slightly lower for the short form of the test. The authors point out that in special education, “almost every third pupil” (p. 105) showed lower test performance in the PC version due to a lack of computer experience [14]. The validity of a digital test format has been supported for other instruments for assessing fluid intelligence as well [27,28].

In 2017, we converted the CFT 20-R in a mobile application suitable for an assessment using smartphones or tablets without the need for observation and instructions by a test administrator. As addressed extensively above, a change in modus can influence test results. Therefore, we cannot assume that the psychometric properties of the online version are identical to those of the PP version. The psychometric properties of a test are generally described in terms of their objectivity, reliability, validity, and norms [29]. Due to the high degree of standardisation of the instructions (played audio file) as well as the calculation of the test results (automatised) of the online version, the objectivity of the online version can be assumed to be at least as good as that of the PP version. Acceptable reliability, validity, and norms cannot be assumed in advance, highlighting the need for an evaluation.

### 1.5. Research Questions and Hypotheses

The aim of the current study was to examine the reliability and validity of the test results of the online CFT 20-R. More specifically, the following research questions formed the basis for evaluating the reliability and construct and criterion validity of the online version:Is the internal consistency of the total score of the online version satisfactory?

Because the test content of the PP and online versions did not differ, we expected that the internal consistency of the online version is comparable to that of the PP version as reported in the manual.
2.To what extent do the test results of the online version, in which intelligence is measured by means of the three subtests–Series, Classifications, and Matrices–support its construct validity?

We expect that the general intelligence as reported in the manual (general fluid intelligence; [14] pp. 16, 32, and 78–79) can be assessed using the three subtests of the online version. We expect this to be reflected in high correlations between the subtest raw scores on the one hand and the total raw score on the other hand, as well as in a good fit for a bifactor model (with item loading on both the subtest factor and the overarching intelligence factor).
3.How strong is the relationship between the test results of the online version and school grades in mathematics and German?

Based on previous findings on the PP version, we expect the correlation to be higher with mathematics grades than with German grades [14]. We expect that the results of the online version predict school grades just as well as the results of the PP version do. As intelligence can predict grades in both mathematics and German [30], this would support the criterion validity of the results.

In addition, the following two questions were raised exploratively. These relate to the comparison between the online and PP versions of the CFT 20-R:4.Do the results of the online and PP versions differ (a) at the level of the subtest and total test score, (b) at the item level, and (c) with respect to the classification into IQ categories commonly used in diagnostic practice?

Previous studies have shown that test modus effects are small in the case of (I) tests using tablets and smartphones, and (II) assessment of figural reasoning in unobserved online tests. Therefore, we expected a high correlation between the test results of the online and PP versions. Due to the study design, in which all participants completed the online version first, the test results are expected to be higher for the PP version than for the online version. We expect this difference to be comparable to differences generally found when evaluating the retest reliability of a test, which are influenced by learning effects of repeated measurements, children’s developmental progress, and measurement error.

5.Does the agreement between the online and PP versions depend on whether the children show (a) below-average intelligence or not, or (b) impulsive behaviour or not?

Based on previous findings on the interaction between the mode effect and performance level (below-average vs. average) in reading, spelling, and mathematics [31,32], we expect a high agreement between both test versions, independent of the child’s intelligence category. Due to a lack of previous research on the influence of impulsive behaviour on mode effects, we do not have an a priori expectation regarding the answer to question 5 (b).

## 2. Materials and Methods

### 2.1. Data Collection

The current study is based on the data from an online study, for which a software company (Meister Cody GmbH, Düsseldorf, Germany) has programmed various tests and questionnaires, including the CFT 20-R, in app format. In order to obtain a representative sample of third and fourth grade children in Germany, 52,734 families with a child in third of fourth grade in the German states of Hesse and Bavaria were invited to participate in the study by letter via the local governments. The letter contained the data to log in to the study’s app via a smartphone or tablet, where families could give informed consent for their participation. After this, the children could complete the CFT 20-R and various achievement tests via the app. Parents were given the following instruction: “You are welcome to support your child if he or she does not understand the instructions. However, we ask you to not provide your child with the answers or solutions to the tasks”. Parents and children additionally filled in questionnaires about internalising and externalising problem behaviour of the child. The tasks for the children were divided in blocks of 30 to 45 min per day on four different days. The CFT 20-R was completed on the first day, just after completing a questionnaire about academic self-concept.

After the online study was finished, a subsample of the participating families was invited for a PP assessment in which the CFT 20-R and the achievement tests were completed in PP form. In Bavaria, families from the city of Munich were invited; in 177 of these families, the child had shown below-average achievement (standardised T-score ≤ 40) in reading, spelling, and/or mathematics, and in 112 families, the child had obtained average achievement. In Hesse, all participating families had the opportunity to participate in the PP assessment. The tests were administered by trained student assistants.

The online study took place in May 2017; the PP assessments took place from June to September 2017 (in Hesse) and from October 2017 to the beginning of January 2018 (in Bavaria). The time interval between the online and PP tests was 108.2 days on average (SD = 63.1; range = 22–240). This means that, at the time of the PP assessments in Bavaria, the children had already moved to the fourth and fifth school grade. The ethics committees of the University Hospital of the Ludwig Maximilian University Munich (project ID: 438-16; date of approval: 25.08.2016) and of the DIPF | Leibniz Institute for Research and Information in Education, Frankfurt am Main (project ID: FoeDises; date of approval: 02.04.2017) reviewed and approved the study.

### 2.2. Sample

In total, 4542 families logged in to the app. In 390 cases, the CFT 20-R was not completed or the data were implausible. We excluded the data of one sibling per pair (*n* = 47; randomly), as well as from children who were below 8.5 years of age (*n* = 5), which is outside the age range for the norms of the CFT 20-R. The total sample contained 4100 children who completed the online CFT 20-R. Of these, 220 children also participated in the PP assessment because their parents responded to the invitation for a PP assessment after completing the online study, as described above (see Section 2.1). In 47.2 % of the total sample and 47.7 % of the PP sample, parents reported that their child had their own smartphone or tablet. Further information about the participants’ familiarity with the use of digital devices was not assessed during the study. Families without their own tablet or smartphone had the opportunity to procure a device for the duration of the study. Table 1 contains the descriptive statistics for both (dependent) samples.

### 2.3. Materials

#### 2.3.1. Intelligence: CFT 20-R Online and PP Versions

Part 1 (short form) of the CFT 20-R was used in the current study. The reliability of Part 1 is 0.92, as reported in the manual (Lienert formula 27; [14]). The time limit was 4 min for the subtests Series and Classifications and 3 min for the subtests Matrices and Topological Conclusions. The subtests consist of 15 items each, apart from Topological Conclusions, which consists of 11 items.

Three of the four subtests were transformed into the online version. For Topological Conclusions, this transformation was not possible due to technical reasons. The children were guided through the app based on a cover story, in which Master Cody (a magician) played the main role. In the PP version, the test administrator gave the instructions with detailed explanations and example items. In the online version, a simplified version of this instruction was spoken by Master Cody, after which the child was invited to answer two example items. To ensure that the child had understood the tasks correctly, the example items were offered again when answered wrongly. The actual test was only started after the child had answered all example items correctly.

Figure 1 shows the presentation of the items in each of the subtests in the two versions. In the online version, the response options in the subtests Series and Matrices were shown below the task, which differed from the PP version, where the response options were shown next to the task. The children were asked to tap the right answer on the screen. The chosen response option was then given a different colour. Correction of an answer or scrolling back to an earlier item was not possible.

#### 2.3.2. School Grades

The parents filled in a questionnaire about background information about the family and the child. One of the questions asked for the child’s school grades in German and mathematics (“What was the grade in the last school report of your child in German/math?”; research question 3).

#### 2.3.3. Impulsivity 

Impulsive behaviour (research question 5b) was assessed during the online study using a questionnaire for parent-reported ADHD-symptoms, which is part of the German diagnostic system for mental disorders in children and adolescents (DISYPS-II; [33]). This questionnaire contains 20 items with a four-point Likert scale (Cronbach’s alpha = 0.82), 4 of which measure impulsivity. The items describe behaviours that may apply to children to varying degrees (e.g., “Often blurts out answers before questions are finished”).

### 2.4. Analyses

Because the children could not be observed during the online assessment, plausibility checks were conducted. More specifically, we excluded data in which the test duration was implausibly short or long or response times were implausibly short (*n* = 314; 7%). The norms of the PP version could not be applied, because the fourth subtest was missing in the online version. Therefore, age and grade norms (third and fourth grade) were developed based on the online sample. For more detailed information about the plausibility checks and norm development, we refer to Visser et al. [34]. 

Because information about exact age was missing for some of the children, we applied the grade norms in the current study. Even though 28 of the 144 children from Hesse and all children from Bavaria had already moved to the next grade in school at the time of the PP assessment, we applied the grade norms appropriate to the grade the child was in during the online study (3rd or 4th) for all children, for better comparability.

For the online version, we determined internal consistency (McDonald’s omega) using a macro for SPSS (version 26.0; IBM Corp.; Armonk, NY, USA) [35]. We evaluated the construct validity based on the correlations between the subtest raw scores and the total score as well as a confirmatory factor analysis (CFA). For the CFA, we specified a bifactor model in MPlus (version 8.4; Muthén & Muthén; Los Angeles, CA, USA) [36], in which the items loaded on both the subtest factors and the overall factor. We used variance-adjusted weighted least squares (WLSMV) as the estimator and the following criteria for judging model fit: root mean square error of approximation (RMSEA) ≤ 0.06 and comparative fit index (CFI) ≥ 0.95 [37]. 

We calculated the correlation with school grades in German and mathematics for the complete sample, and for grades 3 and 4 separately. In addition, we used a linear regression to evaluate the extent to which the test results on the online version could predict school grades.

Because the research design in the current study was not counterbalanced, research questions 4 and 5 could only be evaluated exploratively. To achieve this aim, we created descriptive statistics and correlations for the raw scores on the level of the total test, subtests, and items. Because the time interval between the online and PP assessment was relatively long and differed between the children, we additionally evaluated the influence of this time interval using a regression analysis with the raw score on the PP version as the dependent variable. The raw score on the online version and the number of days between the two assessments were the predictor variables.

To evaluate if the difference in raw score between the PP and online versions differed between children with and without below-average intelligence (IQ in the online version ≤ 85 versus > 85) and between children with different degrees of impulsivity, we applied a repeated-measures ANOVA. We operationalised impulsivity on the basis of the average score on the four items related to impulsivity: 0 = no impulsivity; up to 1 = slight impulsivity; 1 or higher = high impulsivity [33].

To evaluate if the speed–accuracy emphasis (the tendency to focus on speed or accuracy in test-taking) differed between children with different levels of impulsivity, we also used a repeated-measures ANOVA. As recommended by Borter, Troche, and Rammsayer [18], we calculated the speed–accuracy emphasis by calculating the difference between the number of not-answered items and the number of incorrectly answered items. 

## 3. Results

### 3.1. Internal Consistency (Research Question 1)

Based on the data from the online study, we found an internal consistency (McDonald’s omega) of 0.75 for the total CFT 20-R. This is comparable to the value of 0.73 reported in the manual of the PP version. The internal consistency for the subtests was 0.49 (Series), 0.50 (Classifications), and 0.65 (Matrices). The manual of the PP version reports internal consistency values for the three subtests between 0.68 and 0.77 [14].

### 3.2. Construct Validity (Research Question 2)

The correlation between the subtests and the overall score for the online version was 0.75 (Series), 0.77 (Classifications), and 0.82 (Matrices). This corresponds to the values reported in the manual of the PP version between 0.78 and 0.83 ([14], p. 16).

The results of the CFA showed a good model fit for the bifactor model based on the raw scores of the three subtests in the online version (*n* = 4.100): χ2 (900) = 1704.27, *p* < 0.001, RMSEA = 0.015 (90% confidence interval 0.014–0.016), and CFI = 0.95.

### 3.3. Relationship with School Report Grades (Research Question 3)

School grades in Germany range from 1 to 6, with 1 being the best grade. For the total sample, the correlation between the total raw score of the online version and the reversed parent-reported German and mathematics school grades was *r* = 0.29 and 0.32, respectively. When looking at third and fourth grade separately, the correlation coefficients were very comparable (see Table 2) and all significant (*p* < 0.01). In the sample of the online study, the correlation coefficients for the German and mathematics school grades differ from each other significantly only in the sample of children in fourth grade (*z* = −1.73; *p* < 0.05).

Table 2 also shows the correlations as reported in the manual of the PP version [14] between the CFT 20-R raw score and the reversed German school grades. These correlations are all higher than the ones we found in the online study, except for the correlation with the school grade in German in children in third grade, for which the correlation coefficients did not differ between the versions.

The results of the linear regression show that the raw score of the online version significantly predicts school grades in German (Beta = −0.253; *p* < 0.01) and in mathematics (Beta = −0.287; *p* < 0.01). The model explains 6% (German) and 8% (mathematics) of the variance in school grades. Adding the raw score of the PP version as a predictor in the model changes the results, in that the PP raw score significantly predicts school grades in German (Beta = −0.292; *p* < 0.01) and mathematics (Beta = −0.362; *p* < 0.01), but the online raw score no longer does so (Beta = −0.120; *p* = 0.095 for German and Beta = −0.122; *p* = 0.079 for mathematics). This model explains 13% and 19% of the variance, respectively.

### 3.4. Comparison between the Online and PP Versions (Research Question 4)

Table 3 shows the means and standard deviations of the raw scores for both versions. The results of the online version for the sample of 220 children who completed both versions are very comparable to those of the large online sample. The comparison between the two versions is based on the sample of 220 children. The results show clear differences that favour the PP version for the total raw score, the IQ score, as well as the subtest raw scores for Series (large effect size) and Matrices (medium effect size). The raw scores for the two versions on the subtest Classifications do not differ significantly and only show a weak correlation. This is also reflected in the confidence intervals around the mean for both versions, which overlap only for the Classifications subtest.

We repeated the analyses based on the norms that were valid for the grade in which the child was at the time of the PP test. In this case, the norms used were thus not the same for all children (see above). The results show an average IQ for the PP version of 110.6 (*SD* = 13.8; Cohen’s *d* = 0.71). The correlation with the IQ of the online version was *r* = 0.45 (*p* < 0.01). The mean IQ differed significantly between the two versions (*t*(219) = 10.1; *p* < 0.01).

Table 3 shows that the variance in the total scores and in the subtests scores for Series and Matrices is larger in the online version than in the PP version. These differences in variance are statistically significant for the Matrices subtest (likelihood ratio [LR] χ2 = 7.19; *p* = 0.004) and for the total raw score (LR χ2 = 3.99; *p* = 0.023), but not for the Series subtest (LR χ2 = 2.24; *p* = 0.067) or for the IQ score (LR χ2 = 0.93; *p* = 0.167). The differences between the raw scores of the children are thus larger in the online version, especially in the Matrices subtest.

The results of the regression analysis showed that the raw score in the online version significantly predicted the raw score in the PP version (Beta = 0.496; *p* <.01). The time interval between the two assessments did not explain additional variance (Beta = −0.087; *p* = 0.139).

A comparison of the item difficulties between the online and PP version showed relatively large differences for four items in the Series subtest (*p* ≥ 0.2), with the items of the online version being more difficult. These items included all three items in which the correct response option was “a” (first of five response options) as well as one item in which “b” (second of five response options) was the correct answer. For the Classifications subtest, only one item showed a clear difference in difficulty, with the PP version being more difficult. For the Matrices subtest, six of the items were more difficult in the online version and one was more difficult in the PP version. The Appendix A contains a complete overview of the item difficulties.

Finally, we tested to which extent the diagnostically relevant classification in IQ ranges (IQ ≤ 85 versus IQ > 85) corresponded between the two versions, which was the case for 188 of the 220 children (85.5%). Three of these children showed a below-average IQ based on both versions. For 32 children (14.5%) the results of the online version, but not the PP version, showed a below-average IQ. The reverse did not occur, which fits with the higher scores found for the PP version than for the online version.

### 3.5. Group Differences in the Correspondence between the Versions (Research Question 5)

Table 4 shows the test results for children with different levels of intelligence and impulsivity. More specifically, it shows the mean raw score per subgroup, the difference between the raw scores of the two versions of the CFT 20-R, and the correlation between both. Figure 2 visualises these results.

#### 3.5.1. Intelligence Category

The raw score difference between both versions is larger for children with an IQ in the below-average range (IQ ≤ 85; *M*-Online = 17.6; *M*-PP = 25.1; difference = 7.5), compared to children with an IQ in the average range (*M*-Online = 27.5; *M*-PP = 30.2; difference = 2.7; see Table 4). The correlation between the raw scores of the online and PP version is statistically significant for both groups. The results of the repeated-measures ANOVA show a main effect (*F*(1,218) = 131.8, *p* < 0.001), which reflects the difference in test results between the online and PP version, as well as a significant interaction between the raw score and group (*F*(1,218) = 29.1, *p* < 0.001). Figure 2 visualises this interaction: The difference between the groups with below-average versus average intelligence is larger for the online version than for the PP version. Part of the children with a below-average intelligence based on the online version show a clearly higher raw score in the PP version (Figure 2b).

#### 3.5.2. Impulsivity

The mean raw scores (*M*) for the PP version do not differ much between children without (*M* = 29.6) or with slight (*M* = 29.5) or high (*M* = 29.0) impulsivity. In the online version, the raw scores of children with high levels of impulsivity are slightly lower (*M* = 25.1) compared to children without (*M* = 26.0) or with slight (*M* = 26.4) impulsivity (see Table 4). The results of the repeated-measures ANOVA with the impulsivity level as factor showed that these differences are not significant: *F*(2,217) = 0.86, *p* = 0.42 for the main effect and *F*(2,217) = 0.62, *p* = 0.54 for the interaction. This does not change when impulsivity is included in the analysis as a continuous instead of categorical variable: *F*(1,218) = 1.49, *p* = 0.22 for the main effect and *F*(1,218) = 0.46, *p* = 0.50 for the interaction. The correlation between the raw scores of the two versions was significant within each of the three groups and were comparable in strength: the lowest (0.44) and highest (0.59) correlation coefficient did not differ significantly from each other (*z* = −1.34, *p* = 0.09). 

Regarding the speed–accuracy emphasis, we found differences between neither the online and PP versions (*t*(219) = 0.688, *p* = 0.492) nor the groups of children with different levels of impulsivity in the two versions: *F*(2,217) = 0.364, *p* = 0.695 for the main effect and *F*(2,217) = 0.545, *p* = 0.581 for the interaction. The correlation between impulsivity and the speed–accuracy emphasis was not significant for the PP version (*r* = 0.044, *p* = 0.514) and low but significant for the online version (*r* = −0.069, *p* < 0.01). The correlation between the speed–accuracy emphasis and intelligence was significant (*p* < 0.01): *r* = 0.731 for the online version and *r* = 0.378 for the PP version.

## 4. Discussion

In the current study, we evaluated the reliability and validity of the test results on the basis of the CFT 20-R online version for children in the third and fourth grades in Germany (research questions 1 to 3). In addition, we performed an explorative comparison of the results of the online version with those of the PP version (research questions 4 and 5).

More specifically, research questions 1 to 3 concerned the internal consistency, construct validity, and criterion validity of the test results based on the online version. For the total score, the results showed satisfactory internal consistency (α > 0.7; [38]), comparable to the internal consistency of the PP version. For the three subtests, the internal consistency was not satisfactory and was also lower than the values reported in the manual of the PP version [14]. This speaks for the use of the total IQ score.

Our results support the construct validity of test results based on the online version. With respect to the criterion validity, we found a lower correspondence between online test results and school grades in German and mathematics, compared to the values reported in the manual for the PP version. The results do support the expectation that the test results would correspond to school grades in mathematics more than to those in German for the children in fourth grade, but the differences are small. This could be due to the fact that the school grades were based on parent reports, which might be less reliable than school grades that are reported by the school directly. The results of the regression analyses showed that the results of the online version can predict school grades in German and math, but not as well as the results of the PP version can. This could be explained by the fact that school achievements are usually also evaluated on the basis of PP assessments. In addition, the results on the online version could have been influenced by other characteristics of the children, such as their digital competence. Within the scope of the study, the only information available was whether the children had their own tablet or smartphone. A complementary analysis using repeated-measures ANOVA revealed no main effect of owning a tablet or smartphone on the difference in raw scores on the online and PP assessments, *F*(1,218) = 0.012, *p* = 0.912. Furthermore, there was no significant interaction between the raw score difference and owning a tablet or smartphone, *F*(1,218) = 0.637, *p* = 0.426. However, since owning a device does not reflect digital competence, more research is needed to answer the question how digital competence might affect the results on the online version.

Research question 4 concerned possible differences between the online and PP versions with regard to (a) results at subtest and total test level, (b) results at item level, and (c) diagnostically relevant IQ categories (IQ ≤ 85 versus IQ > 85). The raw scores obtained with the PP version were a bit higher, which is not surprising, because all children in the study first completed the online version and then the PP version due to organisational constraints. It is well known that repeated testing leads to practice gains due to a combination of learning effects and development of the children, e.g., [39]. The CFT 20-R manual ([14], S. 51) reports an increase of 3 points in the raw score and 6 to 7 IQ-points for a test interval of 2 to 3 months in ninth grade pupils. The test interval in the current study was 3.5 months on average. Since developmental gains in fluid intelligence are higher in younger than in older children, e.g., [40,41], the identified difference of 3.5 raw score points lies within a range that was to be expected based on repeated testing and developmental gains, even though it was based on three instead of four subtests.

A possible explanation for the larger difference of 12.9 IQ points between the online and PP version, as compared to Weiß [14], could be the demographic characteristics of the sample of the online study. Children from families with a high educational background of the parents were slightly overrepresented and those with a migration background were slightly underrepresented [34]. This could have caused the norms of the online version to be a bit too strict. Another possible explanation could be derived from the norms used for the PP version. More specifically, we used the norms for the third and fourth grade for all children to ensure comparability, even though a part of the sample had already moved on to the fourth and fifth grades, respectively. We verified how the results changed when using the norms of the fourth and fifth grades instead for those children who had already moved grade, which showed a slightly smaller difference of 10 IQ points between the versions. The effect size was comparable to that of the raw score difference.

For the subtest Classifications we did not find a learning effect: the average scores on both versions did not differ. This indicates that this subtest is somewhat easier in the online version compared to the PP version. Another remarkable finding was the larger variance in the scores for the subtest Matrices, as well as in the overall raw test score for the online version compared to the PP version. Possibly, children are more used to assessments in PP format, causing the result of the online assessment to depend to a larger extent on personal characteristics instead of only intelligence. For example, children who have relatively more extensive experience with a smartphone or tablet could have shown better test results. The question why this difference in variance was only found for the subtest Matrices cannot be answered based on the results of the current study.

The manual of the PP version reports retest correlation coefficients for the subtest raw scores in grades 3 to 9 between 0.46 and 0.62. We found lower correlations (0.28–0.38) between the subtests of the two versions. This might be due to a combination of factors. First, the young age of the participants could play a role, because the results of intelligence tests are generally less reliable in young children [42]. Second, the long time interval between the two assessments for some of the children could have played a role. Third, the low correlation could also have been caused by a mode effect, which would hint at a limited comparability between the PP and online versions of the CFT 20-R. 

On the item level, we found differences between the two versions for some of the items, especially within the subtests Series and Matrices. For the Series subtest, these could have been caused by the change in presentation, because especially items for which the first response option was correct were correctly answered less often in the online version. These items were a bit easier in the PP version, possibly due to the presentation of the response options next to the task, causing the right response option to be located very close to the empty box in the task (see Figure 1). In the online version, the response options were located below the tasks, causing the first response option to be located further from the task, compared to the presentation in the PP version. For the subtests Classifications, the item presentation was very comparable between the two versions, which could have caused the high agreement between them. We have not been able to find an explanation for the differences in difficulty found for some of the items in the Matrices subtest. In general, we derive from the findings that separate norms need to be developed and used for an online version of a test if changes in the item presentation are needed.

The final research question focused on possible group differences in agreement between the versions. The results showed the largest differences between the raw scores of the two versions for children with a below-average test result (IQ ≤ 85) in the online assessment. This was also reflected in a relatively low correspondence between the results of the online and PP version in these children, for whom the online version seemed to be a relatively difficult. Maybe a proportion of these children could benefit relatively more from an assessment in PP format. In addition, the learning effect could be larger for these children. 

This needs to be studied in more detail in future research, because especially for children with an IQ in the low average or below-average range there is a high need for reliable and valid intelligence tests. If the online version were to be relatively more difficult for these children compared to the PP version, this would cause a problem with the test fairness, possibly resulting in an increased risk in an underestimation of the intelligence of these children. This could have severe consequences in daily practice, for example in cases of an intelligence assessment in the context of diagnosing an intellectual disability or specific learning disability (see [43]).

We have not found any differences in the score difference between the versions when comparing children with different degrees of impulsive behaviour. In addition, the speed–accuracy emphasis did not differ between the versions or between children with different degrees of impulsivity. These results support the comparability of the test results of both versions and thus the validity of the online version. 

We also did not find clear differences in speed–accuracy emphasis between children with different intelligence levels. However, a significant correlation between the speed–accuracy emphasis and intelligence showed that children with a higher intelligence, on average, work less quickly and make fewer mistakes. This result contrasts to those of earlier studies, which did not find such a correlation [18]. A possible explanation is that, in case of the CFT 20-R, the average number of items that were not answered due to the time limit was very low (0.4 for the online version and 1.0 for the PP version; SD = 1.1). This causes the value for the speed–accuracy emphasis to be influenced very strongly by the number of incorrectly answered items, which could be related to intelligence. In addition, as is the case for the speed–accuracy trade-off [44], the emphasis on speed versus accuracy could be influenced by motivation aspects, causing a lack of correlation on the group level.

### 4.1. Limitations and Future Research

The current study has a couple of limitations, which restrict the interpretation of the results. First, the children made the online CFT 20-R at home without the supervision of a test administrator. Consequently, the assessment and circumstances could not be observed and controlled. Even though the parents were instructed not to help their child by giving the solutions, we cannot be sure that the children worked on the tasks independently, in a focussed manner, and without disturbances from their environment. This was the reason for the extensive plausibility checks, which enabled us to exclude data from clearly unreliable tests and thereby minimize the influence of reliability issues on the study results. The lack of supervision during the online assessment and less-detailed instruction given could, however, be an additional explanation for the higher average performance in the PP assessment, in addition to the retest effect. We expect this influence to be small given the results of studies also in children that showed only small effects of unobserved online formats in assessments, for example for figural reasoning and reading comprehension [22,45]. In addition, online assessments at home may well have advantages in terms of test fairness. Assessment in a familiar environment and without being observed can reduce performance anxiety and thereby positively influence the test performance. This holds potential that has not yet received much attention, especially in diagnosing specific learning disorders.

Second, as we did not have information about the kind of device used by the families in the online study, we could not evaluate possible differences between test results obtained via smartphones and tablets. Such differences cannot be ruled out, because the answer buttons were larger on tablets than on smartphones and the size of the buttons could influence the test performance, e.g., [46,47]. 

Third, the online version contained only three of the four subtests of the CFT 20-R. The subtest Topological Conclusions could not be transferred in an online format due to its complex item presentation. Nevertheless, the results show a relatively high agreement between the total scores of the two versions. In addition, the results of the CFA were satisfactory. This indicates that a sufficiently valid assessment of fluid intelligence is possible on the basis of the three subtests.

Fourth, and importantly, the study design forms a clear limitation, because all children first made the online and then the PP version. A counterbalanced design, which helps to avoid the confounding between the retest effects and version effects, was not feasible in the current study for organisational reasons. In answering the last two research questions, in which we directly compared the results on both versions, we were therefore not able to distinguish between retest and version effects. The results of a descriptive comparison with the retest results reported in the CFT 20-R manual suggest that the raw score differences found in the current study are probably caused by a combination of learning effects and developmental gains of the children. The interpretation of the differences between the test results of children with and without below-average intelligence is also affected by the confounding due to the study design; the results of the other group comparisons are not.

The fifth limitation also relates to the comparison between the online and PP versions: the time interval between the online and PP assessment varied greatly within the sample. Although we did not find any differences depending on the time interval, the study needs to be replicated with a fixed time interval and a counterbalanced design to further validate the conclusions. 

Finally, it should be noted that, in general, the intrinsic characteristics of PP and digital assessment instruments differ. The aim of the present study was to transfer a PP test into a digital format while staying as close as possible to the original version of the test. Possibly, further changes could have been made to the test materials and/or response format in order to obtain a more optimal online CFT 20-R. Future studies could examine which changes should optimally be applied in order to transfer a PP test into a digital format. In addition, future research could focus on assessment instruments that were specifically developed for PP or online testing, respectively, and compare the effects of environmental and personal characteristics on test performance.

### 4.2. Implications for Daily Praxis

Altogether, the results of the current study show that an online assessment of the intelligence of children in third and fourth grade is possible. The online version of the CFT 20-R can reliably and validly measure intelligence, although some differences were found with the PP version, which indicates that separate norms for the online version are clearly appropriate.

The potential to assess intelligence in an online format offers various advantages and will most likely play an increasingly important role in the future. The associated automatised scoring yields a clear time advantage for the person who administers the test and is less prone to errors. Given the fact that children’s living environment is becoming increasingly digital, the ecological validity of the test results is likely to become greater for online assessments, compared to PP assessments, in the future.

## Figures and Tables

**Figure 1 children-09-00512-f001:**
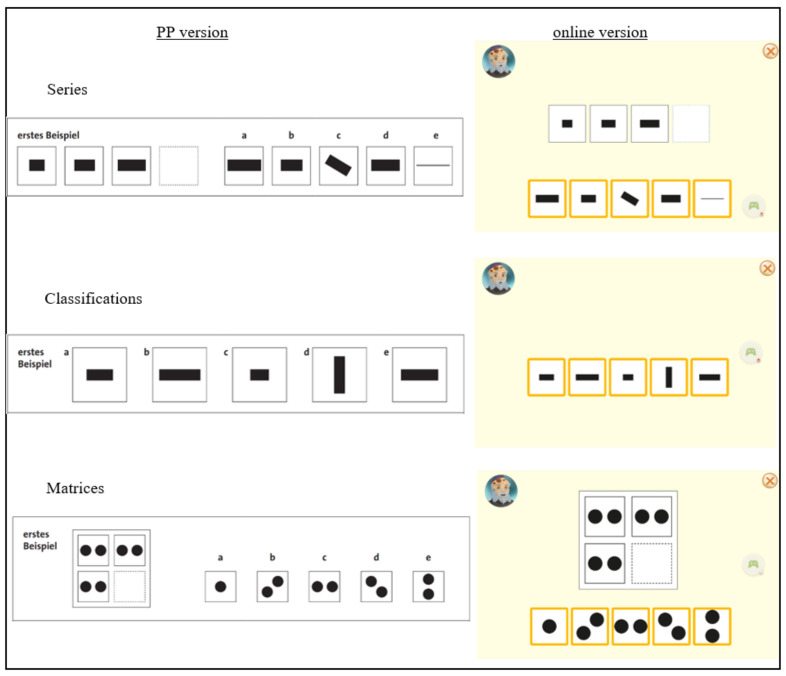
Comparison of the Paper-Pencil (PP) and online version of the CFT 20-R. “erstes Beispiel” = “first example”, which is a practice item including an explanation to the children on what to do. The letters a, b, c, d, and e represent the answer options in the PP-Version, which need to be checked on a separate paper.

**Figure 2 children-09-00512-f002:**
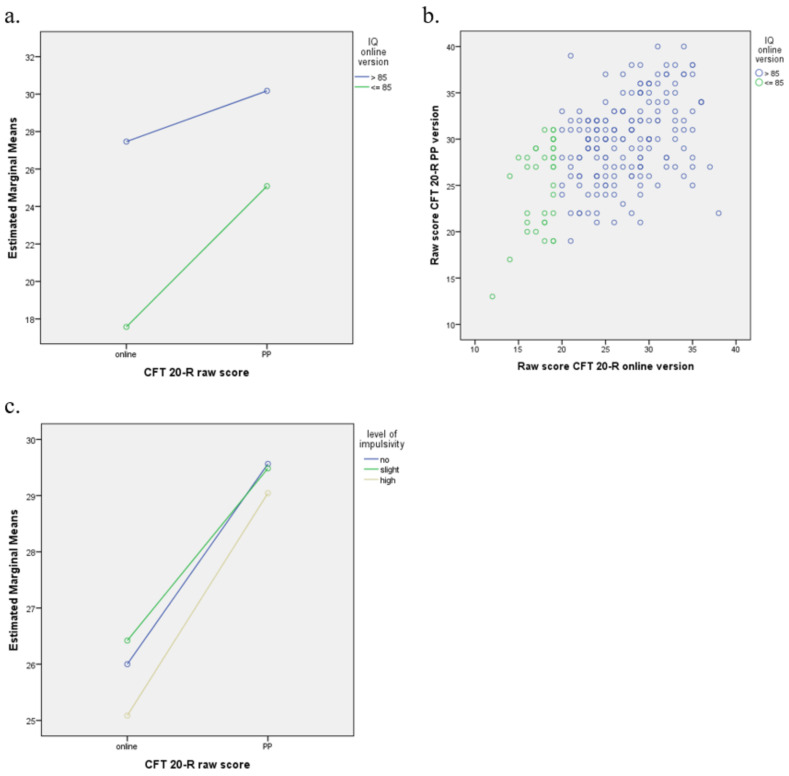
Raw scores for the Paper-Pencil (PP) and online versions of the CFT 20-R for children with different levels of intelligence (line chart (**a**) and scatter plot (**b**)) and impulsivity (line chart (**c**)).

**Table 1 children-09-00512-t001:** Descriptive statistics for the samples of the online and PP assessments.

		**Online** **(*n* = 4100)**	**PP** **(*n* = 220)**
state	Bavaria	3153	76
	Hesse	947	144
grade *	3	1869	66
	4	2231	154
gender	girl	1943	94
	boy	2157	126
age	*M* (*SD*)	9.9 (0.7)	10.0 (0.7)
	range	8.5–12.0	8.8–11.10
days between the tests	*M* (*SD*)	N/A	108.2 (63.2)
	range	N/A	22–240

*Notes.* PP = Paper-Pencil; * grade at the time of the online assessment; N/A = not applicable; *M* = mean; *SD* = standard deviation.

**Table 2 children-09-00512-t002:** Correlations between the CFT 20-R raw scores and school grades in German and mathematics.

	Third Grade		Fourth Grade	
	German	Mathematics	German	Mathematics
Online	0.29 (*n* = 1623)	0.32 (*n* = 1626)	0.29 (*n* = 1939)	0.34 (*n* = 1944)
PP (Weiß, 2006 [14])	0.39 (*n* = 218)	0.46 (*n* = 218)	0.46 (*n* = 218)	0.53 (*n* = 218)
Comparison online vs. PP:				
*z*	−1.56	−2.28	2.80	−3.33
*p*	0.06	0.01	<0.01	<0.01

*Notes.* PP = Paper-Pencil.

**Table 3 children-09-00512-t003:** Raw scores of the CFT 20-R online and PP versions as well as results of their comparisons (correlation; *t*-test).

	Online (*n* = 4100)			Online (*n* = 220)			PP (*n* = 220)			Comparison		
	*M* (*SD*)	95% CI	Range	*M* (*SD*)	95% CI	Range	*M* (*SD*)	95% CI	Range	*r*	*t*	Cohen’s *d*
Series	8.6 (2.1)	8.6–8.7	0–15	8.8(2.1)	8.5–9.1	3–14	10.8(1.9)	10.5–11.0	5–15	0.34 **	−12.7 **	0.99
Classifi-cations	8.4 (2.3)	8.3–8.5	0–15	8.5 (2.3)	8.1–8.8	1–14	8.6 (2.3)	8.3–8.9	1–14	0.28 **	−0.8	0.07
Matrices	8.4 (2.5)	8.3–8.5	0–15	8.7(2.5)	8.3–9.0	2–14	10.0 (2.1)	9.7–10.3	4–15	0.38 **	−7.7 **	0.58
Total raw score (3 subtests)	25.4 (5.4)	25.3–25.6	9–43	25.9 (5.4)	25.2–26.6	12–38	29.4 (4.8)	28.7–30.0	13–40	0.50 **	−10.0 **	0.68
IQ value (grade norms)	100.0 (14.5)	99.4–100.5	69–131	100.6 (14.7)	98.6–102.5	69–131	113.5 (13.8)	111.6–115.3	74–152	0.47 **	13.0 **	0.91

Notes. PP = Paper-Pencil; CI = confidence interval; ** *p* < 0.01; *M* = mean; *SD* = standard deviation.

**Table 4 children-09-00512-t004:** Raw scores of the CFT 20-R online and PP versions for specific subgroups.

		Online *M (SD)*	PP *M (SD)*	Difference *M (SD)*	Correlation *r*
Intelligence range ^a^	≤85 (*n* = 35)	17.6 (1.8)	25.1 (4.7)	7.5 (4.2)	0.45 **
	>85 (*n* = 185)	27.5 (4.3)	30.2 (4.4)	2.7 (4.9)	0.35 **
Impulsivity ^b^	no (*n* = 55)	26.0 (5.2)	29.6 (4.8)	3.6 (5.0)	0.50 **
	slight (*n* = 95)	26.4 (5.8)	29.5 (4.4)	3.1 (5.6)	0.44 **
	high (*n* = 70)	25.1 (4.8)	29.0 (5.3)	4.0 (4.6)	0.59 **

*Notes.* PP = Paper-Pencil; ** *p* < 0.01; *M* = mean; *SD* = standard deviation. ^a^ On the basis of the IQ score from the online version; ^b^ No: average score = 0; slight: 0 > average score < 1; high: average score ≥ 1; average score means the average answer for the four questions related to impulsivity on the questionnaire for parent-reported ADHD-symptoms (4-point Likert scale from 0 to 3).

## Data Availability

The datasets used and/or analysed during the current study are available from the corresponding author on reasonable request.

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
