# Peer review of "Evaluation of an Online Version of the CFT 20-R in Third and Fourth Grade Children"

_children, 2022, doi:10.3390/children9040512_

Round 1

Reviewer 1 Report

This article is an interesting topic to research. 

From the analysis of the theoretical framework section of the article, some conclusions can be deduced that can be quite enlightening and significant:
    *The ideas appear perfectly organized, structured and contextualized, which causes the reader to have a broad overview of the most significant arguments and information handled in that section of the article.
*The main scientific antecedents of the problem under study, both national and international or regional, are continuously mentioned.
*The current situation or situation in which the problem of the study is contextualized and located is also described with great precision.
*The ideas are frequently supported and endorsed by a wide range of high quality and current bibliographic references, although with a periodicity of more than 10 years, in most cases, and complying, at all times, with the guidelines established and marked in the APA regulations.

The purpose of the study is amply described and contextualized throughout the article in language that is very clear and precise. The sample that ended up being part of the study is extensively described and, since it has a fairly representative number of subjects in all the parameters analyzed in the article, it can be considered to have good levels of significance and representativeness, a key element for the final validation of the results derived from the empirical study in question.

The instruments used in the study are extensively described, so that all the instruments that were finally implemented are quite clear, as well as the objective or purpose for which they were used.
The treatment or analysis of the study data is very well documented and complemented with a wide arsenal of tests and statistical procedures, so that it becomes very clear how the data collected as a result of the implementation of the study were analyzed and treated and, therefore, the information that they managed to report to the study.
The results of the study, as they are written, sequenced and ordered, are quite congruent with the main objectives set out in the article. In addition, it is also necessary to highlight that the results described are in line with the most significant methodological approaches established and planned in the study and, therefore, are, for the most part, conveniently organized and written, at least from a narrative, chronological and statistical point of view.
The discussion, for the most part, is characterized by an extensive critical and broadly contextualized analysis of the most significant results of the study, avoiding, as is required by convention and the most elementary rules of ethics and scientific procedure, a repetitive analysis of the main achievements described in the results section of the article.
The conclusions of the study are correctly stated, organized and described, to the point that they end up delimiting, with relative clarity, the scientific space through which the different empirical studies that, in the future, intend to continue with the wake or the path that the present study has left to explore.

However, there is a basic flaw or error when carrying out this study, and that is the desire to transfer a test or content in digital format, just as it is in physical format. Doing so is a mistake, since they are different media, with different intrinsic characteristics, so it makes no sense to do exactly the same thing in two formats that are completely different. 

It would be interesting, therefore, to reflect on this, and for future research, to take into account the necessary changes that must be made in the tests or materials, in order to be able to be transferred to a digital format.

Reviewer 2 Report

The study Sample lacks a more thorough description. For example, what is the participants’ familiarity with the use of digital devices? What is the participants’ familiarity with digital testing? Also, to what extent are we facing a sample that includes participants with no background of the presence of digital in the family? Does the study predict the effect of this background or, on the other hand, does it put these children in a situation of double harm? In fact, in the discussion of the results, the authors explicitly mention that the results in the online version may be biased by children's characteristics such as digital competence (p. 13, lines 476-477). Therefore, the sample should be described in detail in this regard and the In the Discussion section it is recommended that the resulting implications can be explicitly discussed.

Regarding Method, what were the methodological procedures for constituting the subsample for the comparative analysis between the online version (n= 220) and the PP version (n= 220)? This needs explicit clarification in the ms.
